# A Worldwide Analysis of Adipose-Derived Stem Cells and Stromal Vascular Fraction in Orthopedics: Current Evidence and Applications

**DOI:** 10.3390/jcm12144719

**Published:** 2023-07-17

**Authors:** Robert Ossendorff, Alessandra Menon, Frank A. Schildberg, Pietro S. Randelli, Sebastian Scheidt, Christof Burger, Dieter C. Wirtz, Davide Cucchi

**Affiliations:** 1Department of Orthopaedics and Trauma Surgery, University Hospital Bonn, Venusberg-Campus 1, 53127 Bonn, Germanydieter.wirtz@ukbonn.de (D.C.W.);; 2Laboratory of Applied Biomechanics, Department of Biomedical Sciences for Health, Università degli Studi di Milano, Via Mangiagalli 31, 20133 Milan, Italy; 3U.O.C. 1° Clinica Ortopedica, ASST Gaetano Pini-CTO, Piazza Cardinal Ferrari 1, 20122 Milan, Italy; 4Dipartimento di Scienze Cliniche e di Comunità, Scuola di Specializzazione in Statistica Sanitaria e Biometria, Università degli Studi di Milano, Via Mangiagalli 31, 20133 Milan, Italy; 5Research Center for Adult and Pediatric Rheumatic Diseases (RECAP-RD), Department of Biomedical Sciences for Health, Università degli Studi di Milano, Via Mangiagalli 31, 20133 Milan, Italy

**Keywords:** osteoarthritis, adipose tissue, mesenchymal stem, stromal cells, adipose-derived stem, stromal vascular fraction, microfragmented adipose tissue, regenerative medicine

## Abstract

The biological enhancement of tissue regeneration and healing is an appealing perspective in orthopedics. We aimed to conduct a systematic review to describe the global distribution of studies investigating the use of adipose tissue derivates in orthopedics and to provide information on their quality and on the products available. The quality of the included studies was assessed using the modified Coleman Methodology Score (mCMS) and the Cochrane risk-of-bias tool for randomized trials. Eighty-two studies were included, with a total of 3594 patients treated. In total, 70% of the studies investigated the treatment of knee disorders, predominantly osteoarthritis; 26% of all studies dealt with expanded adipose-derived stem/stromal cells (ADSCs), 72% of which had stromal vascular fraction (SVF); 70% described the injection of adipose tissue derivates into the affected site; and 24% described arthroscopies with the addition of adipose tissue derivates. The mean mCMS for all studies was 51.7 ± 21.4 points, with a significantly higher score for the studies dealing with expanded ADSCs compared to those dealing with SVF (*p* = 0.0027). Our analysis shows high heterogeneity in terms of the types of performed procedures as well as the choice and processing of adipose tissue derivates.

## 1. Introduction

Strategies to enhance tissue regeneration and healing are appealing perspectives for orthopedic surgeons, aiming to reduce pain in degenerative joint conditions, postpone replacement procedures, restore painless function in tendon pathologies, and to improve results after reconstructive approaches. Efficient and effective healing is necessary to restore normal function after acute injuries or degenerative changes to cartilage and tendon tissue, but these tissues have limited natural healing capacity; therefore, biological enhancements, such as growth factors, stem cells, bioactive scaffolds, and tissue engineering approaches could play a crucial role by helping to overcome the limited regeneration potential of specific tissues where natural healing is slow or challenging. Furthermore, by tailoring treatments to individual patient needs, such as using autologous stem cells or customized tissue engineering scaffolds, a personalized approach can be achieved, with improved outcomes and reduced risks of adverse reactions [1,2,3,4,5,6].

New biological solutions have been proposed to achieve these goals. First, growth factors were proposed as a possible solution to stimulate tissue regeneration, followed by platelet-rich plasma (PRP), which was recognized as a possible autologous source of these molecules. Finally, mesenchymal stem/stromal cells (MSCs) gained popularity in regenerative medicine for their potential to enhance tissue healing. MSCs are believed to enhance tissue healing through the stimulation of local cells via the paracrine mechanism and anti-inflammatory and/or immunomodulatory activity, thus creating a suitable microenvironment for tissue repair and regeneration [7,8,9,10,11,12,13,14,15]. In recent years, the adipose tissue and its stromal vascular fraction (SVF) have been demonstrated to provide an easily accessible source for these cells and modulatory substances, with promising results in preclinical models [1]. Considering the low harvest morbidity for adipose tissue and its relative abundance in easily accessible regions of the human body, clinical publications describing the treatment of numerous orthopedic conditions with adipose tissue derivates have flourished. Furthermore, future applications of adipose tissue derivates in combination with bioengineering innovation, such as 3D bioprinting and computational simulation, hold great potential to advance regenerative medicine for osteoarthritis and tissue repair [16,17].

Lipoaspirate or tissue biopsy allow the effective and safe harvest of adipose tissue from the subcutaneous layers, which can then be processed in different ways to obtain specific therapeutic products [18]. These products can be distinguished in three major groups: expanded MSCs, enzymatically extracted SVF, and mechanically extracted SVF (microfragmented or microfractured adipose tissue). Expanded MSCs are obtained via the enzymatic digestion of adipose tissue and are cultured to obtain a well-characterized product with a high and pure concentration of MSCs. Non-expanded adipose tissue derivates contain a cocktail of MSCs, platelets, immune cells, cytokines, and growth factors, defined as SVF, and can be obtained via enzymatic or mechanical methods from unprocessed fat tissue [19]. However, higher regulative demands and costs currently limit the clinical application of expanded MSCs to a few specialized centers. Conversely, the popularity of non-expanded adipose tissue derivates has increased due to their high cost-effectiveness and the possibility of a point-of care application, although the cell yield and characterization has not yet been investigated for most available products. Nonetheless, clinical applications have flourished across the globe in non-specialized settings, leading to a multitude of clinical studies, not always following strict methodological rules. As a result, many publications dealing with different adipose tissue derivates and using different extraction protocols and commercially available devices are available, with publication quality ranging from rigorous clinical trials to case series of questionable scientific value. No current analysis of the quality of the volume of published literature has been performed.

The quality assessment of publications is important to advance scientific knowledge and its practical applications. In particular, the evaluation of methodologic quality is necessary to determine the scientific validity and robustness of research studies by evaluating the study design, methodology, data collection, analysis, and interpretation. Such an assessment has not yet been performed for studies dealing with human-adipose-derived stem/stromal cells (ADSCs) or SVF (including microfragmented adipose tissue) applications to ensure proper research is conducted using appropriate methods to provide meaningful and reliable results. We aimed to conduct a systematic review to analyze the global distribution of studies investigating the use of adipose tissue derivatives in orthopedics, to assess the quality thereof and provide information on the available products.

## 2. Materials and Methods

The Preferred Reporting Items for Systematic reviews and Meta-Analysis statement (PRISMA) was followed as a guideline for the study [20]. This study did not require ethical committee approval. The review was registered on the PROSPERO database (Registration number: CRD42022339795).

### 2.1. Search Strategy

To analyze the use of adipose tissue derivates in orthopedic surgery, a systematic review of the PubMed/Ovid Medline and EMBASE electronic databases was performed. The search string was constructed with the aid of an experienced librarian with expertise in electronic searches, and the search was performed on 25 May 2022. Two reviewers independently applied the predefined eligibility and inclusion criteria to the articles. The references of all the fully assessed articles and relevant review papers were also hand-searched to identify additional articles.

### 2.2. Exclusion and Inclusion Criteria and Study Selection

Randomized controlled trials and prospective cohort studies (level I and II studies) were included; retrospective comparative trials (level III studies) and therapeutic case series (level IV studies) were also considered for inclusion. Reviews, meta-analyses, expert opinions, and editorial pieces were excluded. Animal studies, in vitro studies, and biomechanical studies on human cadaver specimens were also excluded.

Studies enrolling human subjects of all ages receiving treatments of orthopedic pathologies with the application of adipose tissue derivates (unprocessed adipose tissue, non-expanded or expanded ADSCs, SVF, or microfragmented adipose tissue) were eligible for inclusion. The treatment of rheumatoid conditions, diseases of the hand, and applications in plastic surgery, as well as wound healing enhancement protocols were excluded. Studies presenting results of different therapeutical approaches were excluded unless it was possible to identify and isolate data from the subgroup of patients who received the application of adipose tissue derivates.

### 2.3. Quality Assessment

All the studies that met the final inclusion criteria were individually assessed for quality by two independent reviewers, first by assigning a level of evidence according to the recommendations by Marx et al. [21]. Second, the presence of national or international registration as a clinical trial was documented. Furthermore, methodologic quality was assessed to determine the scientific validity and robustness of research studies by evaluating the study design, methodology, data collection, analysis, and interpretation. For this purpose, the quality of all studies was evaluated using the modified Coleman Methodology Score (mCMS). This instrument is utilized to evaluate methodological quality in studies reporting surgical outcomes after patellar tendinopathy. Subsequent, modified versions of the CMS have been proposed to improve the analysis of the quality of studies on different anatomical districts or procedures [22,23,24,25,26].

Methodological quality was evaluated for each paper and graded as excellent (>85 points), good (70–84 points), fair (55–69 points), and poor (<55 points). Grouping, depending on adipose tissue preparation and its processing, was subsequently performed. Finally, the Cochrane risk-of-bias tool for randomized trials (Version 2) was used to assess the risk of bias of Level I and II randomized controlled trials [27]. This tool is widely used to assess the risk of bias in individual studies included in systematic reviews.

### 2.4. Data Extraction, Collected Variables, Grouping, and Analysis

Information regarding authors, journal and year of publication, study design and quality of evidence, country where the study was conducted, anatomical district of application, treated pathology, intervention, and specific characteristics of the adipose tissue derivate used were extracted and entered into a spreadsheet for analysis. In particular, the source of the adipose cells; the setting of the processing (specialized laboratory or point-of-care); the methods of extraction, if expansion and characterization were performed; and if a commercial device was used were considered relevant variables. The included articles were then categorized depending on the country the trial was performed in, the anatomical area of application, the type of adipose tissue preparation, and tissue processing (site, device, extraction, expansion, and characterization).

### 2.5. Statistical Analysis

Statistical analysis was performed using GraphPad Prism v. 6.0 software (GraphPad Software Inc., Boston, MA, USA) and Microsoft Excel Version 2306 (Microsoft Corporation, Redmond, WA, USA). The Shapiro–Wilk normality test was used to evaluate the normal distribution of the sample. Dichotomous variables were expressed in numbers of cases and frequencies; their differences were tested using the chi-square test or the Fisher’s exact test. Continuous variables were expressed as median and interquartile range (first and third quartiles) or mean ± standard deviation, as appropriate. The between-group differences for continuous variables were evaluated with the unpaired Student *t* test or Mann–Whitney test, according to the characteristics of the data distribution evaluated using the Shapiro–Wilk normality test. For all analyses, the significance level was set at *p* < 0.05.

## 3. Results

### 3.1. Review Process and Included Studies

The database search identified 5038 studies. After the removal of duplicates and title and abstract screening, 297 articles were selected. The full-text assessment with an additional hand search of references identified 82 studies that were included in this review (Figure 1).

A trend of an increasing number of publications was observed over the last few years, stagnating after 2019 (Figure 2). This could be related to the COVID-19 pandemic.

A total of 3594 patients were treated with adipose tissue derivates in the included studies.

### 3.2. Worldwide Distribution

Most of the included studies were performed in Asia, with South Korea being the leading country, followed by the United States and Italy; with 42 national studies, these three countries account for over the half of all publications on this topic. Only four multi-national studies (all Europe-based) were published (Figure 3).

Most of the procedures were performed in Europe (European Union (EU) and the United Kingdom (UK)) (2589 patients) followed by Asia (747) and the USA (180). A higher number of patients per study were enrolled in European studies (25.5 [15.50–77.25]) compared to those in Asia (18.00 [12.00–30.00]) and the USA (13.50 [4.75–28.25]), with no statistically significant difference. It should, however, be noted that, among the European-based studies, the largest (1128 patients) is a case series involving different arthritic joints (61.0% knee, 33.7% hip, and 5.3% other joints) of very low methodological quality [28].

### 3.3. Anatomical Area and Treated Pathologies

In total, 57 out of the 82 included studies were specifically dedicated to procedures around the knee (Figure 4). The number of enrolled patients per study was significantly higher for knees compared to others (knee: 20.00 [15.50–25.25]; other areas: 10.00 [1.00–21.5]; *p* = 0.0015).

Most of the available studies investigated the treatment of knee disorders, mainly knee osteoarthritis. Nevertheless, no uniform indication for treatment was encountered, with included Kellgren–Lawrence osteoarthritis grades ranging from 1 to 4. Besides knee osteoarthritis, one study investigated the treatment of juvenile osteochondritis dissecans of the patella [29]; one, patellar tendinopathy [30]; and one, ACL tears [31]. With 3183 treated patients, the knee was confirmed to be the most studied joint in terms of the number of recorded procedures. The Western Ontario and McMaster Universities Arthritis Index (WOMAC), the Knee Injury and Osteoarthritis Outcome Score (KOOS), a Visual Analog Scale, and the 36-Item Short Form Survey (SF-36) were the most frequently used outcome measures.

Regarding the hip joint, osteoarthritis was an indication for treatment [32,33], together with acetabular cartilage delamination in femoroacetabular impingement [34] and osteonecrosis of the femoral head [35,36]. Ankle osteoarthritis [37,38] and Achilles tendinopathy [39,40] were treated in two publications each, and a case report presented the results of treatment of an unstable osteochondral lesion of the talus [41]. Around the shoulder joint, regenerative approaches involving adipose tissue derivates were all directed to rotator cuff pathologies, with two studies dealing with full thickness tears [42,43], three with partial-thickness tears [44,45,46], and one with shoulder pain caused by refractory rotator cuff disease in wheelchair users [47]. The treatment of recalcitrant lateral elbow pain was the topic of three included studies involving the elbow [48,49,50]. Two included studies involved the spine and addressed degenerative disease and chronic discogenic low back pain [51,52]. Four studies reported on the application of adipose tissue derivates in multiple joints, mainly including osteoarthritis patients treated for knee and hip symptoms [28,53,54,55].

### 3.4. Procedures

Almost three-quarters of the included studies focused on injections of adipose tissue derivates into the affected joint or tendon, either alone or combined with PRP or hyaluronic acid (Figure 5).

Arthroscopic procedures with associated injections of adipose tissue derivates were reported in 24% of the cases; half of these were sham procedures (the debridement of arthritic joints, synovectomy, and partial meniscectomy for degenerative lesions), and half included microfractures, ACL reconstructions, and rotator cuff repairs. In two papers, open procedures augmented by the injection of adipose tissue derivates were described (subtalar arthrodesis [38] and high tibial osteotomy [56]).

### 3.5. Adipose Tissue Derivate Types

The preferred source for tissue harvest was autogenic subcutaneous fat. The abdomen, flanks, buttocks, and thighs were the mentioned sources of subcutaneous fat tissue. In three studies, autologous fat tissue was harvested during the surgical procedure from a local reservoir as follows: the infrapatellar fat pad in two studies (25 and 30 patients, respectively [57,58]) and the peritrochanteric area in one (17 patients [34]). The use of allogenic ADSCs was reported in four studies, for a total of 108 patients [38,48,59,60]. Figure 6 summarizes the distribution of graft sources in the included studies.

Approximately one-quarter (26%) of the included studies dealt with expanded ADSCs, whereas the rest investigated the effect of SVF (including microfragmented adipose tissue); no significant differences in the number of patients per study in these two groups were found (ADSCs: 18.00 [12.00–24.00]; SVF: 20.00 [6.00–35.00]; *p* > 0.05). One study described the use of unprocessed fat tissue (Figure 7).

Of the 60 studies (72%) dealing with SVF, 30 used enzymatic and 30 non-enzymatic extraction methods. A characterization of the used cells was performed in less than half of the included studies (39%) (Appendix A).

### 3.6. Adipose Tissue Processing

Two-thirds of the included studies presented the application of adipose tissue derivates as a point-of-care procedure without cell expansion. The commercial products used in these 55 studies are reported in Figure 8, whereas relevant information on the site and type of processing is summarized in Appendix A.

### 3.7. Quality of Evidence Appraisal

A level of evidence was assigned to each study according to the recommendations by Marx et al. [21] as follows: 15.9% of the included studies were classified as Level I and 19.5% as Level II trials. More than the half of the included studies were case reports or case series (Figure 9).

National or international clinical trial registration was obtained in 35% of the included studies. The level of evidence of studies using expanded ADSCs was significantly higher than that of studies using SVF (ADSCs: 2.00 [2.00–4.00]; SVF: 4.00 [2.00–4.00]; *p* < 0.001).

The mean mCMS for all included studies was 51.7 ± 21.4 points, with a significantly higher score for the studies dealing with expanded ADSCs compared to those dealing with SVF (mean mCMS: 63.9 ± 17.8 and 47.7 ± 21.4, respectively, *p* = 0.0027). A wide variability in the methodological quality of the studies was encountered, with most of the studies ranking in the “poor” or “fair” categories (76.8%). Reasons for low scores were mainly retrospective study designs, a lack of clearly reported diagnostic and inclusion criteria, as well as the low number of included patients and short follow-up. The average study quality in terms of mCMS was higher for Asian and USA-based studies (Asia: 56.2 ± 23.0, *p* = 0.0366) compared to the EU and the UK (USA: 55.9 ± 25.0, p: n.s.), when compared to the EU and the UK (EU and UK: 47.2 ± 17.6; Australia: 43.8 ± 18.8; Argentina: 39.5 ± 31.8). This trend was maintained when considering the percentage of studies with excellent or good (mCMS ≥ 70) methodological quality (Asia: 37.1%; USA: 30%; EU and UK: 6.7%; Australia: 20%; Argentina: 0%). The results of the Cochrane risk-of-bias analysis for Level I and II randomized studies are reported in Figure 10.

Considering these studies with a high level of evidence, the literature does not adequately investigate many of the aforementioned applications of adipose tissue derivates, such as symptomatic knee osteoarthritis, in particular, with Kellgren–Lawrence grade 2 and 3 (with controversial results in some randomized controlled trials) [58,61,62,63,64,65,66,67], partial- and full-thickness rotator cuff tears [42,45], and Achilles tendinopathy [39]. Promising results have been obtained in Level II trials on the treatment of lateral recalcitrant elbow pain [48] and degenerative disc disease [51,52] as well as in the augmentation of open-wedge high tibial osteotomy [56], yet these should be confirmed in Level I trials. All other previously listed treatments (osteochondritis dissecans of the patella, patellar tendinopathy, ACL tears, hip osteoarthritis, osteonecrosis of the femoral head, and ankle osteoarthritis) do not encounter any sound support in the currently available literature. These treatments should, therefore, be considered as experimental and only administered within adequately designed and monitored research protocols.

## 4. Discussion

Our systematic analysis of publications dealing with adipose tissue derivates in orthopedics showed high heterogeneity in terms of types of performed procedures as well as the choice and processing of adipose tissue derivates. Although high-quality studies have been produced, a high volume of publications show low methodological quality, especially for studies dealing with SVF derivates.

Only a few clinical applications were investigated with adequate high-level publications of symptomatic knee osteoarthritis, rotator cuff tears, Achilles tendinopathy, and subtalar arthrodesis. Promising results have been obtained for the treatment of lateral recalcitrant elbow pain and degenerative disc disease and in the augmentation of open-wedge high tibial osteotomy, yet these studies are biased by a lower level of evidence. No other application is currently supported by adequate high-quality literature, and care should be taken in the use thereof. Through permissive regulations, South Korea, the United States, and Italy are the countries in which the most studies were conducted.

### 4.1. Historical Remarks on MSC Sources for Orthopedic Applications

The term MSCs was introduced in 1991 to identify a group of multipotent, adult stem cells, which have the potential to differentiate into various types of mesenchymal tissues. These cells are characterized by a definite behavior in culture, a subset of surface markers, and differentiating abilities [70,71,72]. As opposed to the initial belief, more recent research demonstrates that, rather than through direct contribution to tissue regeneration through differentiation, MSCs move to sites of tissue injury and direct the regenerative response through the paracrine secretion of bioactive, trophic, and immunomodulatory agents [7,8,9,10,11,12,13,14]. These cells then contribute to create a suitable microenvironment for tissue repair, rather than repairing the tissue themselves; this suggests that an alternative acronym, “Medicinal Signaling Cells”, be considered more suitable than the actual activity of these cells [73].

Bone marrow has historically provided the classical source of MSCs used for orthopedic applications. This site is easy to access and provides a high number of MSCs, without related donor-site morbidity. During the last decade, several distinct populations of MSCs have been isolated from specific periarticular tissues (such as synovium, ligaments, tendons, and bursa), suggesting the possibility of perioperative extraction to avoid the violation of the iliac crest. Nevertheless, the clinical application of these cells remains limited due to the relatively small amount of tissue available for harvest and the need for expansion in a dedicated facility which might affect the functional phenotype of the cellular product [74,75,76,77,78,79,80,81]. Similar to periarticular tissues, adipose tissue also hosts local, resident MSCs. Furthermore, adipose tissue is frequently found in abundant quantities in easily surgically accessible regions of the human body. This triggered the development of techniques to obtain autologous adipose tissue samples with minimal patient morbidity for either processing in a dedicated laboratory or with point-of-care devices.

### 4.2. Adipose Tissue and Its Derivates for Regenerative Medicine Applications

ADSCs represent an appealing perspective for orthopedic surgeons, demonstrating the possibility of the enhancement of tissue regeneration and healing in preclinical studies [1,82]. Adipose tissue can be harvested in abundance from the subcutaneous tissue via lipoaspirate or tissue biopsy [18]. There are also a large variety of possibilities to process adipose tissue to a specific therapeutic product, which can be distinguished between (1) expanded MSC products and (2) non-expanded adipose tissue derivates, including enzymatically and mechanically obtained SVF.

ADSCs can be extracted from adipose tissue via enzymatic digestion and are plastic adherent when maintained in cell culture. Expanded ADSCs products are often characterized in clinical studies with typical surface markers, CD105, CD73, and CD90, as well as the absence (<2%) of CD45, CD34, CD14, CD19, and HLA-DR, and provide a high and pure concentration of therapeutic MSCs. Higher regulative demands and costs result in a lower number of worldwide studies being performed, but these studies are of a higher quality (Figure 7). Nevertheless, there is high heterogeneity in study protocols with different harvesting and cell culture methods, doses, repetitions, carriers, application methods, and co-treatments in musculoskeletal regenerative therapy [83]. Moreover, a limited or poor comprehension of the mesenchymal therapeutic source material may actually constitute the most salient rate-limiting step in achieving positive clinical outcomes. While adipose cells do indeed constitute an often-chosen resource, it is without question that other cell populations engender far greater plasticity—and thereby far greater therapeutic potential. Numerous basic science experiments point to sources other than adipose tissue as possessing both a marked ability to dedifferentiate as well as transdifferentiate. Moreover, the expansion of source cells, particularly those amplified in a monolayer, often delimits such cells from providing therapeutic utility. It may also well be the currently incomplete comprehension of the operative variables that actually influences the outcomes stem cells provide in the setting of senescent and degenerative morbidities. 

Non-expanded adipose tissue derivates can be extracted from unprocessed fat tissue and contain a cocktail of MSCs, platelets, immune cells, cytokines, and growth factors, defined as SVF [19]. Cell isolation can be performed using enzymatic methods (collagenases or proteases) or, alternatively, the SVF can be separated from the connective tissue via mechanical methods such as centrifugation or microfragmentation. This last method leads to the production of microfragmented adipose tissue [19,84], which can be directly used in clinical application after tissue harvest, while the enzymatic SVF production time is dependent upon enzyme digestion. It is important to consider that different harvest and isolation methods can also affect cell viability and the functional phenotype of MSCs [18,85]. These effects are specifically reported after enzymatic digestion.

Alternatively, non-enzymatic processing leads to a lower cell yield, which also depends on the harvesting method [86]. Recently, non-expanded adipose tissue derivates have gained popularity in clinical applications (Figure 7). The advantages of these products are high cost-effectiveness and the possibility of a point-of care application without extensive cell culture. Nevertheless, the fraction of MSCs in these products is low compared to expanded MSC products, and there is a lack of systematic characterization and regulative guidelines [87]. Therefore, it remains questionable if these therapeutical non-expanded adipose tissue approaches can directly be compared with expanded and characterized ADSC products. Figure 11 summarizes the relevant features of the adipose tissue derivatives investigated in our review.

### 4.3. Limitations

This systematic analysis of publications dealing with human applications of adipose tissue derivates has some limitations. The first is directly related to the type and quality of the included studies, which leads to high heterogeneity. Different procedures for different conditions as well as different methods and measures are reported, which make the analysis of the outcomes more complex. This heterogeneity arises from the recent introduction of these applications in orthopedics, and the current lack of clear recommendations on their use, which leads to the flourishing of low-quality publications. This heterogeneity, especially for the methodologies of highlighted studies through quality appraisal, derives from the complexity of this unexplored research area and highlights the need for an improvement in methodological quality for in vivo studies dealing with adipose tissue derivates. A further limitation is related to the characteristics of the adipose cells used in the different studies. First, a characterization of the cells used was performed in less than half of the included studies (39%); furthermore, patient-related factors such as obesity and hormonal status may affect the characteristics of the harvested cells beyond their immunological phenotype, thus leading to slightly different effects when used in clinical applications. Finally, differences in national regulations and reimbursement systems limit the availability of some of the experimental procedures and the commercially available products in some countries, restricting researchers to a limited portfolio of adipose tissue derivates. The lack of regulations is reflected in the lack of a consensus on the use of such products in the international orthopedic community.

## 5. Conclusions

This systematic analysis of publications dealing with adipose tissue derivates in orthopedics shows a high heterogeneity in terms of types of performed procedures as well as the choice and processing of adipose tissue derivates. Although high-quality studies have been produced, a high volume of publications show low methodological quality, especially for studies dealing with SVF and microfragmented adipose tissue. Only few clinical applications were investigated with adequate high-level publications, mostly dealing with symptomatic knee osteoarthritis. Promising results with lower levels of evidence have been obtained for the treatment of lateral recalcitrant elbow pain and degenerative disc disease and in the augmentation of open-wedge high tibial osteotomy, yet further research is required to fill the current high-quality literature gap. No other application is currently supported by adequate high-quality literature, which means that extreme care and appropriate monitoring are advised.

## Figures and Tables

**Figure 1 jcm-12-04719-f001:**
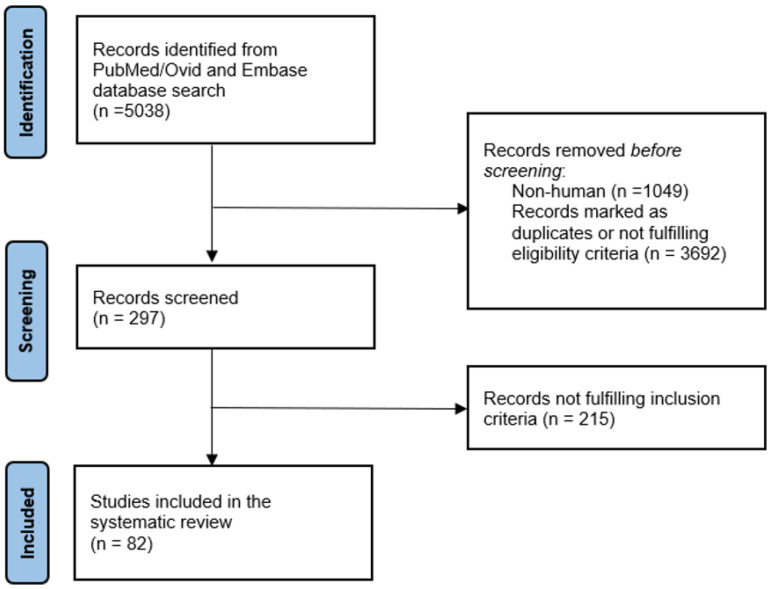
A flow chart showing the selection of publications for the systematic review.

**Figure 2 jcm-12-04719-f002:**
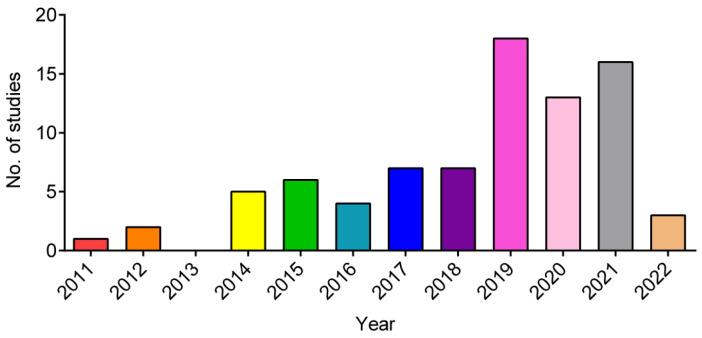
Publication trend over the last 12 years, showing a constant increase in the number of publications dealing with adipose-derived stem/stromal cells (ADSCs) or stromal vascular fraction (SVF) (including microfragmented adipose tissue) applications. For 2022, studies were included up to the review date (March 2022).

**Figure 3 jcm-12-04719-f003:**
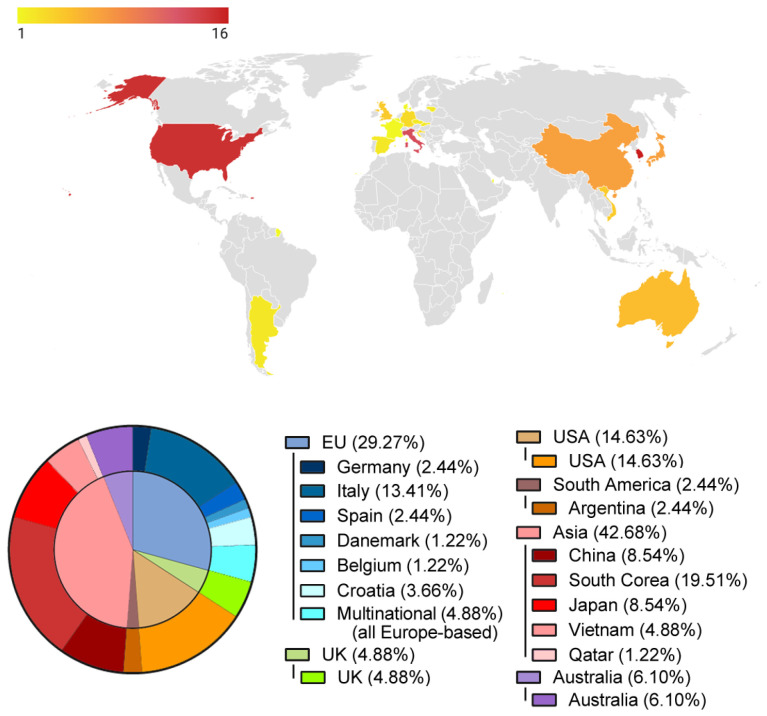
Geographical distribution of the included studies. EU, European Union; UK, United Kingdom; USA, United States of America.

**Figure 4 jcm-12-04719-f004:**
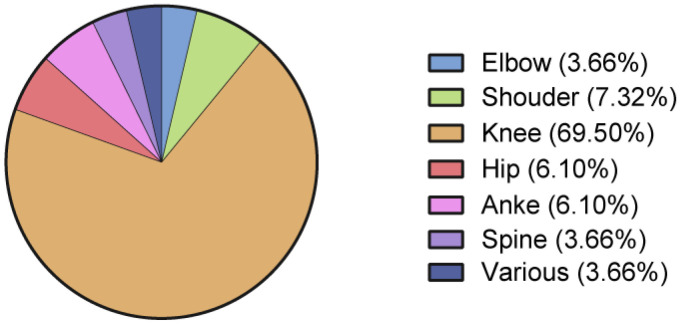
Anatomical areas of procedures performed in the included studies.

**Figure 5 jcm-12-04719-f005:**
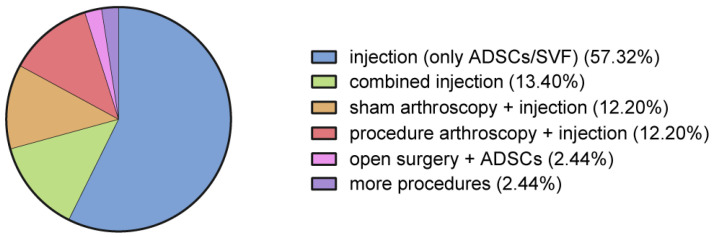
Procedures performed in the included studies.

**Figure 6 jcm-12-04719-f006:**
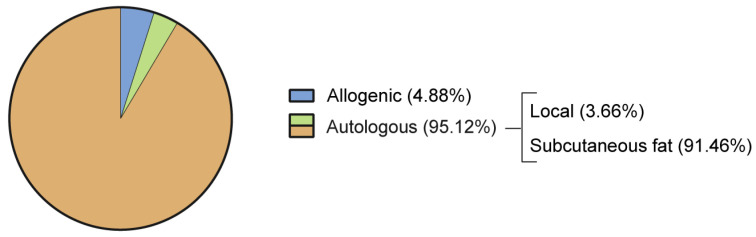
Source of adipose tissue considered in the included studies.

**Figure 7 jcm-12-04719-f007:**
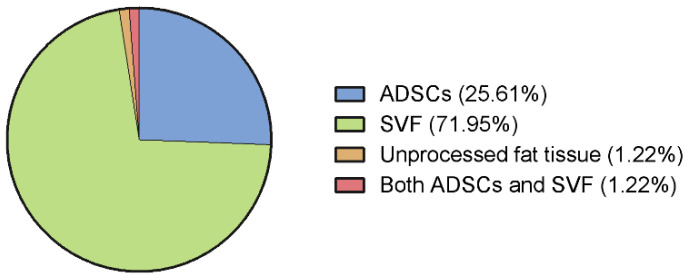
Adipose tissue derivate types used in the included studies.

**Figure 8 jcm-12-04719-f008:**
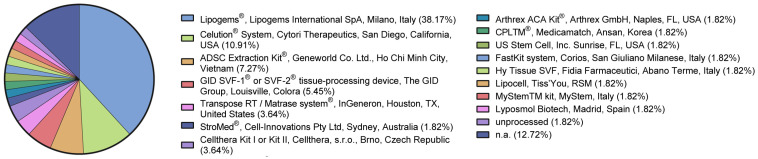
Commercial products used in the included studies.

**Figure 9 jcm-12-04719-f009:**
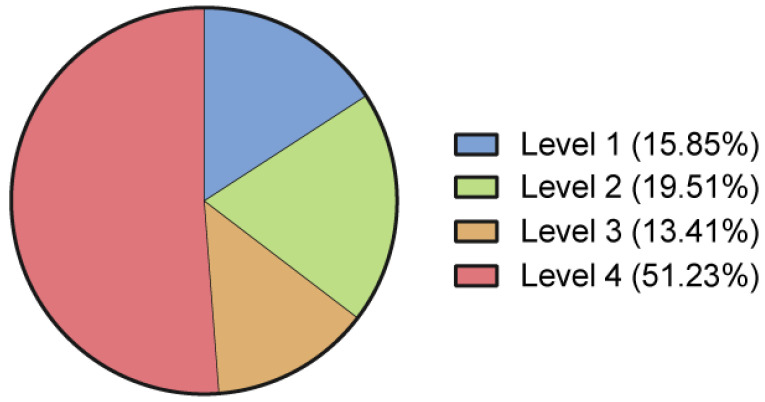
Level of evidence of the included studies, according to Marx et al. [21].

**Figure 10 jcm-12-04719-f010:**
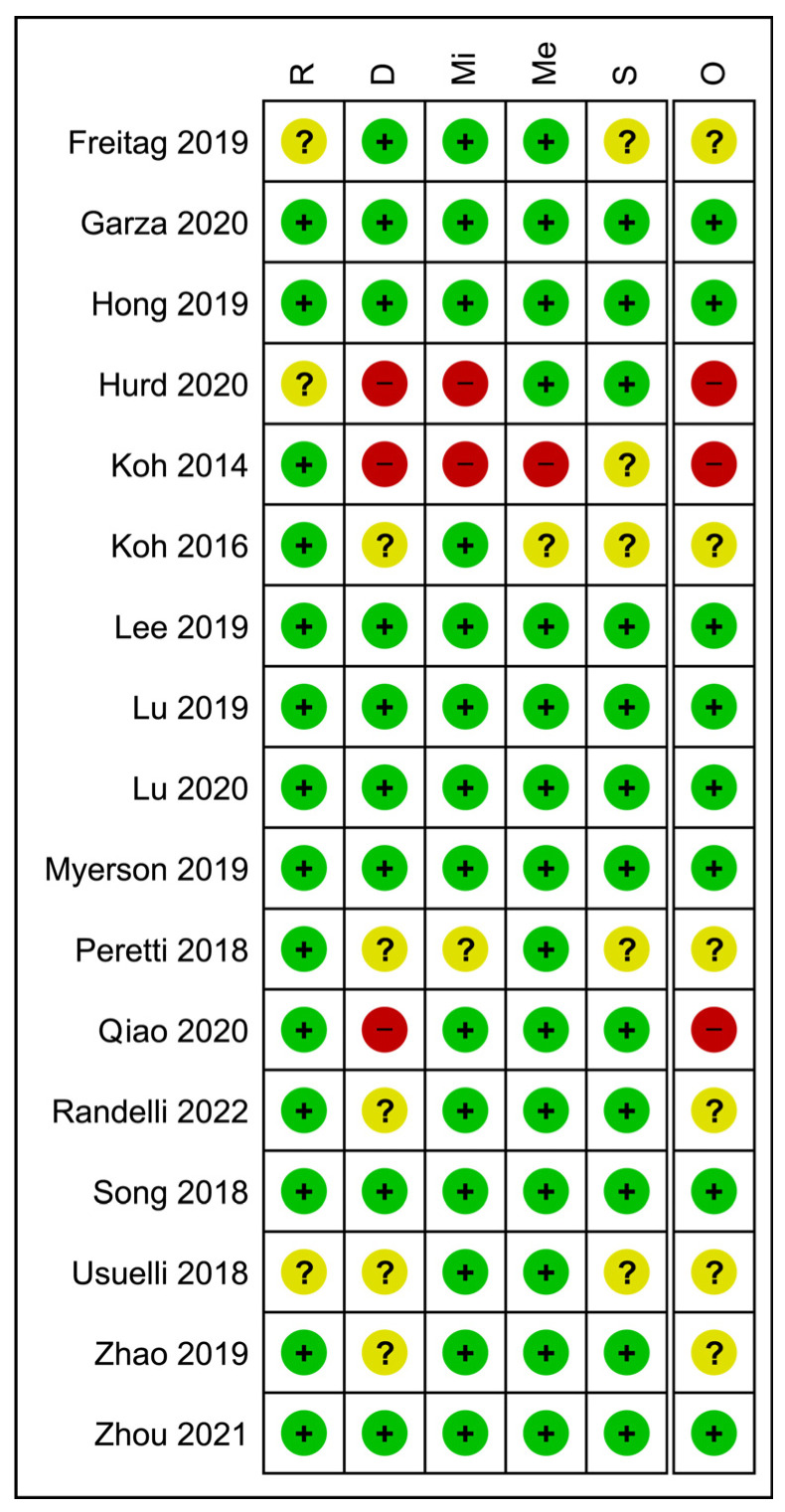
Cochrane risk-of-bias analysis for the included Level I and II randomized studies. R, bias arising from the randomization process; D, bias due to deviations from intended interventions; Mi, bias due to missing outcome data; Me, bias in measurement of the outcome; S, bias in selection of the reported result; O, overall risk of bias; green: low risk of bias; yellow: moderate risk of bias; red: high risk of bias [38,39,42,45,56,58,59,60,61,62,63,64,65,66,67,68,69].

**Figure 11 jcm-12-04719-f011:**
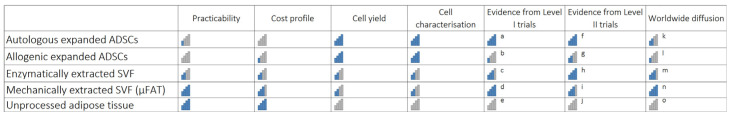
Summary of the relevant features of the adipose tissue derivatives investigated in this systematic review. a: [63,64,65,66]; b: [38]; c: [45,61,67]; d: [39,42,58,62]; e: no Level I trials; f: [52,56,68,88,89]; g: [48,59,60]; h: [51,69,90,91,92]; i: [47,93,94,95]; j: no Level II trials; k: ARG, AUS, CHN, FRA, JPN, KOR, QAT, USA; l: CHN, KOR, USA; m: AUS, CHN, CZE, ESP, GER, JPN, KOR, VTN, USA; n: ARG, AUS, BEL, CHN, DNK, GER, GBR, HVR, ITA, USA; o: USA.

## Data Availability

Not applicable.

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
