# Peer review of "A Worldwide Analysis of Adipose-Derived Stem Cells and Stromal Vascular Fraction in Orthopedics: Current Evidence and Applications"

_jcm, 2023, doi:10.3390/jcm12144719_

Round 1

Reviewer 1 Report

Dear Editor ,

This is with reference to Systematic Review 1 Adipose Derived Stem/ Stromal Cells, Stromal Vascular Frac- 2 tion and Micro-Fragmented Adipose Tissue Applications for 3 Osteoarthritis and in Orthopaedics: A Worldwide Analysis of 4 Current Evidence 5 Robert Ossendorff 1*, Alessandra Menon 2*, Frank Alexander Schildberg 1 , Pietro S. Randelli 2 , Sebastian Scheidt 1 , 6 Christof Burger 1 , Dieter C. Wirtz 1 , Davide Cucchi 1,sent to me for my review.

Comments :

Osteoarthritis and allied Musculoskaeletal diseases and are debilitating and impact the health of  millions across the  globe. This, necessiates  application of new strategies to combat and also minimize the usuage  of NSAID(s) drugs which causes several side effects.

Although surgical interventions seem promising ,nevertheless poses for long term complications effecting the mobility and limits their application. Keeping in view of these limitations  the existing  systemic review submitted  does account for new therapeutic strategies to  address several targets pertaining to musculo skeletal diseases ( 2.3/Fig 4 ) .

Specific comments:

In Materials and Methods  the authors have given a list if Inclusion  and Exclusion criteria for the studies selected in this review, which has been peer reviewed by two experts and considered for the study. They have enrolled human subjects of all age groups for adipose tissue derivates including processed and non -expanded microfragments adipose.

How did they miss the key point in their study in not excluding  post menopausal, high BMI Men and women as Obesity predisposes to several pathophysiological complications including diabetes, portray low grade inflammation and with low immunity leading to an array of dysfunctions insitu ( Madira et al 2012). Also, high BMI results in hyperplasia  and hypertrophy of adipocytes resulting in altered dynamics of adipose tissue vis a vis not suitable for adipose derived transplantation work.

The statistical design should have been more elaborate and should have included these methods too like  - KOOS (Knee Osteoarthritis outcome score), NPRS (Numeric pain rating scale) and SAPS (Short asessment of patient satisfaction) for key statistical methods extensively used for pain management using  Non invasive/ cell therapy approach in addition to the listed ones in the review.

As indicated in 2.6(Quality of evidence apraisal  only 15% classified as level 1 clinical trail using application of adipose tissue derivative as point of care procedure without cell expansion. This point is not clear.

Further,  the average mCMS shows higher score with expanded ADSCs as compared to SVF. However due to the large variation encountered, most of the studies rank in poor categories and this underscores their interpretation  to highlight the application of ADSCs .

It has been well documented that adipose derived MSCs can be obtained in pure population. However it has to be noted that adipose derived MSCs carry the memory  commitment for adipose has due to its inherent multipotent function  and thus more studies should have been included authenticating its application in osteoarthrits and allied complication using the preclinical data.   

Further the systemic review does not arrived at solution at the end of the manuscript, too ambigious and no clear cut strategies illustrated to address different types of musculoskeletal diseases.  in the submitted manuscript.

This review also does not touch upon the 3D bioprinting approach using adipose derived MSCs - Bonemarrow derived MSCs (Autologous or allogenic) to treat AVNs spinal injury and osteoarthritis which is emerging as a promising bioengineering therapeutic approach in the field of more specifically osteoarthritis 

Overall the review is not very informative with lots of gaps to fill in with more review articles justifying their title.

Author Response

Reviewer 1:

Osteoarthritis and allied Musculoskaeletal diseases and are debilitating and impact the health of  millions across the  globe. This, necessiates  application of new strategies to combat and also minimize the usuage  of NSAID(s) drugs which causes several side effects.

Although surgical interventions seem promising ,nevertheless poses for long term complications effecting the mobility and limits their application. Keeping in view of these limitations  the existing  systemic review submitted  does account for new therapeutic strategies to  address several targets pertaining to musculo skeletal diseases ( 2.3/Fig 4 ) .

We thank this reviewer for the appreciation of our work, which presents an updated review of one of the new strategies available to deal with osteoarthritis and other orthopaedic conditions. The precious and constructive critics to the paper have been appreciated and we modified the manuscript according to the suggestions of this reviewer, and we believe this helped ameliorating the quality of the paper.

Specific comments:

In Materials and Methods  the authors have given a list if Inclusion  and Exclusion criteria for the studies selected in this review, which has been peer reviewed by two experts and considered for the study. They have enrolled human subjects of all age groups for adipose tissue derivates including processed and non -expanded microfragments adipose.

How did they miss the key point in their study in not excluding  post menopausal, high BMI Men and women as Obesity predisposes to several pathophysiological complications including diabetes, portray low grade inflammation and with low immunity leading to an array of dysfunctions insitu ( Madira et al 2012). Also, high BMI results in hyperplasia  and hypertrophy of adipocytes resulting in altered dynamics of adipose tissue vis a vis not suitable for adipose derived transplantation work.

We appreciate this precious comment on the role of body weight and hormones in the development of osteoarthritis. Body weight plays indeed a significant role in the development and progression of osteoarthritis acting on multiple pathways; first, excess body weight places increased mechanical stress on the joints, especially weight-bearing joints like the knees and hips; this added pressure can accelerate the wear and tear of the joint cartilage, leading to the development of osteoarthritis. Furthermore, adipose tissue produces inflammatory cytokines which can trigger low-grade inflammation and hormones, such as leptin, that can impact joint health. Postmenopausal women are particularly susceptible to osteoarthritis due to hormonal changes and the decline in estrogen levels, which affects bone density. The current paper is a systematic review of previously published studies. Being these studies mainly on patients affected by osteoarthritis, the coexistence of this disease with adiposity and older age in females is frequent; the choice to include or exclude these patients was made by the primary investigators and cannot be influenced by the authors of the review. It is not possible to perform a selective inclusion only of studies on patients with osteoarthritis and without adiposity and excluding postmenopausal women, because these criteria are not routinely reported in the published studies. Nevertheless, we acknowledge this as a limitation of the included studies (and therefore of our review) and mentioned this in the limitations section.

The statistical design should have been more elaborate and should have included these methods too like  - KOOS (Knee Osteoarthritis outcome score), NPRS (Numeric pain rating scale) and SAPS (Short asessment of patient satisfaction) for key statistical methods extensively used for pain management using  Non invasive/ cell therapy approach in addition to the listed ones in the review.

We appreciate these comments on the role of the collection of postoprocedural outcomes. Indeed , the WOMAC VAS, KOOS and the  SF-36. This has been mentioned in the results section.

As indicated in 2.6(Quality of evidence apraisal  only 15% classified as level 1 clinical trail using application of adipose tissue derivative as point of care procedure without cell expansion. This point is not clear.

This paragraph was rephrased to improve clarity

Further,  the average mCMS shows higher score with expanded ADSCs as compared to SVF. However due to the large variation encountered, most of the studies rank in poor categories and this underscores their interpretation  to highlight the application of ADSCs .

Indeed, studies with expanded ADSCs included in this review have a higher level of evidence and score better in the mCMS score than those with SVF. Caution on interpretation of the results was indicated as suggested  in the limitation section.

It has been well documented that adipose derived MSCs can be obtained in pure population. However it has to be noted that adipose derived MSCs carry the memory  commitment for adipose has due to its inherent multipotent function  and thus more studies should have been included authenticating its application in osteoarthrits and allied complication using the preclinical data.

This systematic review was focused on clinical application of ADSCs and their derivates. Preclinical data on the potential of MSCs is available but preclinical studies were excluded from this systematic review. Nevertheless, we appreciate a lot the comment of this reviewer and improved the introduction and the discussion adding relevant recent information derived from preclinical studies.   

Further the systemic review does not arrived at solution at the end of the manuscript, too ambigious and no clear cut strategies illustrated to address different types of musculoskeletal diseases.  in the submitted manuscript.

The lack of a clear suggestion or guideline on the application of ADSCs and their derivates at the end of this manuscript reflects the lack of such consensus in the international orthopedic community. This has been acknowledged in the limitation section of the manuscript.

This review also does not touch upon the 3D bioprinting approach using adipose derived MSCs - Bonemarrow derived MSCs (Autologous or allogenic) to treat AVNs spinal injury and osteoarthritis which is emerging as a promising bioengineering therapeutic approach in the field of more specifically osteoarthritis 

The 3D bioprinting approach utilizing adipose-derived mesenchymal stem cells (ADMSCs) offers promising prospects in tissue engineering and regenerative medicine. ADMSCs, obtained from adipose tissue, are an abundant and easily accessible source of stem cells that can differentiate into various cell types, making them ideal for bioprinting applications. By combining 3D printing technology with ADMSCs, complex structures and tissues can be fabricated, providing a platform for personalized medicine and organ transplantation; herewith 3D bioprinting could enable the development of functional tissues that can mimic the native architecture and properties of the target tissue, promoting better integration and regeneration This innovative approach holds great potential for advancements in regenerative medicine for osteoarthritis and tissue repair, but has no clinical applications yet, therefore no studies including 3D bioprinting ADSCs could be included. The potential of this technology has been added to the introduction.

Overall the review is not very informative with lots of gaps to fill in with more review articles justifying their title.

The title was modified, also following a suggestion from Reviewer 5

Reviewer 2 Report

The systematic analysis shows a high heterogeneity in terms of types of performed procedures as well as choice and processing of adipose tissue derivates. Some comments given for MAJOR REVIEW as follows.

1.      Line 39, please explain more specific the research progress.

2.      Line 81-87 for searching strategy, please use Scopus, Web of Science, and PubMed as three main data source.

3.      Please make sure the authors have been followed PRISMA 2020.

4.      In line 154-157 for figure 2, please diving more clear explanation in trends of this figure.

5.      Line 182-208, please make it as the sentence rather than point by point as presented.

6.      What is the novel bought by the authors in the current submission? Its works have been widely discussed in the past. Nothing something really new in the present form. The lack of a novel seems to make the present submission like to replication/modified work. The authors need to detail their novelty in the introduction section. It is a major concern for rejecting this paper.

7.      I am do not understand for choosing the pie chart in most of the manuscript. Any reasons? It looks monotonous.

8.      Please explain related about computational simulation/in silico effort as potential area to develop orthopaedic and human healthcare. Some previous related literature needs to incorporated to support the explanation as follows, doi: 10.3390/ma16093298, 10.1038/s41598-023-30725-6, and 10.3390/ma14247554

-

Author Response

The systematic analysis shows a high heterogeneity in terms of types of performed procedures as well as choice and processing of adipose tissue derivates. Some comments given for MAJOR REVIEW as follows.

We appreciate the comments of this reviewer, who focused mainly on the methodological aspects of our work. We improved the methods section as suggested and think these suggestions helped ameliorating the quality of the manuscript.

  1. Line 39, please explain more specific the research progress.

The research progress over the last years has been specified as suggested.

  1. Line 81-87 for searching strategy, please use Scopus, Web of Science, and PubMed as three main data source.

A professional university librarian specialized on systematic reviews performed this review using the aforementioned databases, which are considered necessary and sufficient for a comprehensive review. Scopus searches focus on abstracts and citations, while a search in Embase provides additional insights from its structured full-text indexing. Since Scopus does not use Emtree to facilitate synonym mapping and hierarchical searches, it may retrieve significantly fewer results than Embase, therefore the EMBASE database was preferred.

  1. Please make sure the authors have been followed PRISMA 2020.

The PRISMA 2020 statement was followed to identify, select, appraise, and synthesise the included studies. The PRISMA 2020 checklist was used to verify the completeness of the review and can be submitted to the editorial office upon request. Figure 1 (flowchart) follows the PRISMA 2020 guidelines

  1. In line 154-157 for figure 2, please diving more clear explanation in trends of this figure.

The trend of the data reported in figure 2 was commented as suggested

  1. Line 182-208, please make it as the sentence rather than point by point as presented.

This paragraph was revised as suggested removing bullet points and transforming them in sentences.

  1. What is the novel bought by the authors in the current submission? Its works have been widely discussed in the past. Nothing something really new in the present form. The lack of a novel seems to make the present submission like to replication/modified work. The authors need to detail their novelty in the introduction section. It is a major concern for rejecting this paper.

We thank this reviewer for permitting us to better highlight this aspect of novelty in our publication: this is the first publication analyzing the methodological quality of the available studies dealing with the application of adipose tissue derivates in orthopedics. No publications of such kind exist up to date and this aspect of novelty was specified in the introduction. The authors do not have any similar publication reviewing results of clinical studies on ADSCs or SVF which could be a duplicated publication.

  1. I am do not understand for choosing the pie chart in most of the manuscript. Any reasons? It looks monotonous.

Pie charts are indicated for visualizing data that represents parts of a whole or the distribution of a categorical variable. They are in particular indicated for proportional representations, where they effectively display proportions or percentages of different categories within a dataset. In this paper, pie charts were used where a proportional representation was required.

  1. Please explain related about computational simulation/in silico effort as potential area to develop orthopaedic and human healthcare. Some previous related literature needs to incorporated to support the explanation as follows, doi: 10.3390/ma16093298, 10.1038/s41598-023-30725-6,  and 10.3390/ma14247554 

Some paragraphs and recent relevant literature regarding computational simulation in stem cells therapy have been added to the introduction.

Reviewer 3 Report

1.      Line 39, the authors should refer the related literature for this purpose, please used the suggested: The Effect of Tortuosity on Permeability of Porous Scaffold. Biomedicines 2023, 11, 427. https://doi.org/10.3390/biomedicines11020427

2.      Line 87, what it is mean by “hand searched”?

3.      The justification of “(R.O., D.C.).” does not needed, delete it.

4.      The plan of library words choose in the performed systematic review needs to explain.

5.      Line 106, The mean of “quality assemenet” should be given to avoid the bias mean.

6.      Line 128, difference between ADSCs or SVF is not given?

7.      Line 134, GraphPad Prism is not commonly used? I am first time hear for that. Why not choose others?

-

Author Response

Reviewer 3:

  1. Line 39, the authors should refer the related literature for this purpose, please used the suggested: The Effect of Tortuosity on Permeability of Porous Scaffold. Biomedicines 2023, 11, 427. https://doi.org/10.3390/biomedicines11020427

The suggested reference was added 

  1. Line 87, what it is mean by “hand searched”?

By hand searched is intended the non-automatized search of reference through, for example, reading the references of relevant review papers

  1. The justification of “(R.O., D.C.).” does not needed, delete it.

The justification was removed, as suggested

  1. The plan of library words choose in the performed systematic review needs to explain.

The review details are registered on the accessible PROSPERO Database

  1. Line 106, The mean of “quality assemenet” should be given to avoid the bias mean.

The assessment of the methodologic quality, necessary to determine the scientific validity and robustness of research studies by evaluating the study design, methodology, data collection, analysis, and interpretation, was better specified as suggested

  1. Line 128, difference between ADSCs or SVF is not given?

Comparisons between ADSCs and SVF have been specified in the next section (grouping and analysis)

  1. Line 134, GraphPad Prism is not commonly used? I am first time hear for that. Why not choose others?

GraphPad Prism is a statistical analysis and graphing software widely used in the scientific and research community. It provides a comprehensive platform for data organization, analysis, and visualization, primarily in the fields of life sciences, biomedical research, and social sciences.

Reviewer 4 Report

As a rapidly-evolving therapeutic methodology, the deployment of mesenchymal stem cells constitutes a fascinating and potentially revolutionary approach across multiple armamentaria, including refractory orthopedic pathologies, as articulated in the submitted work.   Particularly in the setting of senescent populations, complex comorbidities can leave those affected with loss of function coupled with ever-increasing pain.  Adipose derived, mesenchymal tissue constitutes an intriguing repository in ameliorating such patients.    Thus, a systematic analysis of the current state of the field should hold the interest of investigators and clinicians alike.  

Here, the Authors selected 297 published articles and 82 studies where various forms of adipose-derived cells were utilized.  Interestingly, in line 151 the Authors observe that after an increasing publication trend, stagnation was observed after 2019.   Perhaps COVID played a role in the noted stagnation. (?)

Authors go on to describe tissue sources, pathologies addressed and outcomes achieved.  From lines 264 to 277 Authors note poor to fair outcomes, with poor inclusion criteria, low patient numbers per study and short follow up listed as causative factors.  High heterogeneity is described as the most important finding.   

Here however, the Reviewer wonders if perhaps a limited or poor comprehension of the mesenchymal therapeutic source material may actually constitute the most salient rate limiting step in achieving positive clinical outcomes.  While adipose cells do indeed constitute an often chosen resource, it is without question that other cell populations engender far greater plasticity -- and thereby far greater therapeutic potential.  Numerous basic science experiments point to sources other than adipose tissue  as possessing both a marked ability to dedifferentiate as well as transdifferentiate.   Moreover, the expansion of source cells, particularly those amplified in a monolayer, often delimits such cells from providing therapeutic utility.  

Absent these critically relevant touch-points the casual reviewer of Author's manuscript may well be left with an incomplete comprehension of the operative variables that actually influence the outcomes stem cells provide in the setting of senescent and degenerative morbidities.   To build in useful context, the Reviewer believes Authors must thoughtfully articulate how and why  all stem cells are not created, nor do they perform, equally.  

  Minor grammatical fine tuning required.  

Author Response

As a rapidly-evolving therapeutic methodology, the deployment of mesenchymal stem cells constitutes a fascinating and potentially revolutionary approach across multiple armamentaria, including refractory orthopedic pathologies, as articulated in the submitted work.   Particularly in the setting of senescent populations, complex comorbidities can leave those affected with loss of function coupled with ever-increasing pain.  Adipose derived, mesenchymal tissue constitutes an intriguing repository in ameliorating such patients.    Thus, a systematic analysis of the current state of the field should hold the interest of investigators and clinicians alike.  

Thank you for this comment. We appreciate these suggestions.

Here, the Authors selected 297 published articles and 82 studies where various forms of adipose-derived cells were utilized.  Interestingly, in line 151 the Authors observe that after an increasing publication trend, stagnation was observed after 2019.   Perhaps COVID played a role in the noted stagnation. (?)

Thank you for these considerations. We included them in the manuscript.

Authors go on to describe tissue sources, pathologies addressed and outcomes achieved.  From lines 264 to 277 Authors note poor to fair outcomes, with poor inclusion criteria, low patient numbers per study and short follow up listed as causative factors.  High heterogeneity is described as the most important finding.   

That is correct. Heteogeneity is one of the major findings of this review which limit the comparison of the studies.

Here however, the Reviewer wonders if perhaps a limited or poor comprehension of the mesenchymal therapeutic source material may actually constitute the most salient rate limiting step in achieving positive clinical outcomes.  While adipose cells do indeed constitute an often chosen resource, it is without question that other cell populations engender far greater plasticity -- and thereby far greater therapeutic potential.  Numerous basic science experiments point to sources other than adipose tissue  as possessing both a marked ability to dedifferentiate as well as transdifferentiate.   Moreover, the expansion of source cells, particularly those amplified in a monolayer, often delimits such cells from providing therapeutic utility.  

Thank you for these considerations. We included them in the discussion.

Absent these critically relevant touch-points the casual reviewer of Author's manuscript may well be left with an incomplete comprehension of the operative variables that actually influence the outcomes stem cells provide in the setting of senescent and degenerative morbidities.   To build in useful context, the Reviewer believes Authors must thoughtfully articulate how and why  all stem cells are not created, nor do they perform, equally.  

Thank you for this comment. We added these considerations in the discussion. For a standardization it is important to consider the therapeutic potential of stem cells in senescence and degenerative morbidities. This might influence the outcome in clinical studies.

Reviewer 5 Report

1.      The title could be more concise and specific. Consider revising it to reflect the main focus of the study, such as "A Worldwide Analysis of Adipose-Derived Stem Cells and Stromal Vascular Fraction in Orthopaedics: Current Evidence and Applications for Osteoarthritis."

2.      In the introduction, provide more context and background information on why the biological enhancement of tissue regeneration and healing is important in orthopaedics. This will help readers understand the significance of the study.

3.      Clearly state the objective of the review in the introduction. Instead of saying "This review aims to describe the global distribution of studies," specify the specific goals, such as "The objective of this review is to analyze the global distribution of studies investigating the use of adipose tissue derivatives in orthopaedics, assess their quality, and provide information on the available products."

4.      Provide a clear methodology section that describes the search strategy used for the systematic review. Include details such as the inclusion and exclusion criteria, databases searched, keywords used, and any other relevant information.

5.      When mentioning the assessment of study quality, specify how the modified Coleman Methodology Score (mCMS) and the Cochrane risk-of-bias tool for randomized trials were applied. Provide a brief explanation of these tools and how they were used to assess the quality of included studies.

6.      Instead of just stating the overall average mCMS score, consider providing a breakdown of the scores for the studies dealing with expanded ADSCs and SVF separately. This will give readers a better understanding of the quality of studies in each subgroup.

7.      In the results section, provide a summary of the main findings in a structured manner. For example, you could present the distribution of studies by pathology (knee disorders vs. others), the distribution of studies by type of adipose tissue derivative used (ADSCs vs. SVF), and the distribution of treatment methods (injection vs. arthroscopy).

8.      When discussing the high heterogeneity in terms of procedures and choice of adipose tissue derivatives, provide some insights or explanations for this heterogeneity. Are there any trends or patterns observed in the studies? This will help readers understand the diversity in the field.

9.      Consider discussing the limitations of the study, such as any potential biases in the included studies, the generalizability of the findings, and any other limitations that may affect the interpretation of the results.

10.  Provide a clear conclusion that summarizes the main findings of the review and their implications for the field of orthopaedics. Additionally, consider suggesting future research directions based on the identified gaps and limitations.

11.  Proofread the manuscript for grammar, punctuation, and formatting errors to ensure clarity and readability.

Requires grammar checking of the entire manuscript.

Author Response

Reviewer 5:

We thank this reviewer for the precious and constructive critics to the paper; we revised the manuscript modifying the title and adding information to the introduction and discussion, which contributed to an important improvement in the quality of the manuscript.

1. The title could be more concise and specific. Consider revising it to reflect the main focus of the study, such as "A Worldwide Analysis of Adipose-Derived Stem Cells and Stromal Vascular Fraction in Orthopaedics: Current Evidence and Applications for Osteoarthritis."

Thank you for this valuable suggestion, we changed the title accordingly

2. In the introduction, provide more context and background information on why the biological enhancement of tissue regeneration and healing is important in orthopaedics. This will help readers understand the significance of the study.

The introduction was expanded adding information both on the background for biological enhancement of tissue healing and on the future perspective of these technologies

3. Clearly state the objective of the review in the introduction. Instead of saying "This review aims to describe the global distribution of studies," specify the specific goals, such as "The objective of this review is to analyze the global distribution of studies investigating the use of adipose tissue derivatives in orthopaedics, assess their quality, and provide information on the available products."

The end of the introduction was modified to improve clarity following this suggestion

4. Provide a clear methodology section that describes the search strategy used for the systematic review. Include details such as the inclusion and exclusion criteria, databases searched, keywords used, and any other relevant information.

The materials and methods section was ameliorated to include the necessary information.

5. When mentioning the assessment of study quality, specify how the modified Coleman Methodology Score (mCMS) and the Cochrane risk-of-bias tool for randomized trials were applied. Provide a brief explanation of these tools and how they were used to assess the quality of included studies.

More information on the mCMS and on the Cochrane risk-of-bias tool were provided in the methods section to improve understanding of these analyses

6. Instead of just stating the overall average mCMS score, consider providing a breakdown of the scores for the studies dealing with expanded ADSCs and SVF separately. This will give readers a better understanding of the quality of studies in each subgroup.

We thank this reviewer for the valuable suggestion. In the methods section, it was better specified that grouping depending on adipose tissue preparation and its processing was performed. The results of this analysis is presented in the results section.

7. In the results section, provide a summary of the main findings in a structured manner. For example, you could present the distribution of studies by pathology (knee disorders vs. others), the distribution of studies by type of adipose tissue derivative used (ADSCs vs. SVF), and the distribution of treatment methods (injection vs. arthroscopy).

We appreciate this suggestion. The results section was structured with the following subheadings:

Review process and included studies
Worldwide distribution
Anatomical districts and pathologies treated
Procedures
Adipose tissue derivate types
Adipose tissue processing
Quality of Evidence appraisal

8. When discussing the high heterogeneity in terms of procedures and choice of adipose tissue derivatives, provide some insights or explanations for this heterogeneity. Are there any trends or patterns observed in the studies? This will help readers understand the diversity in the field.

A section “limitations” was added at the end of the discussion, which deals with the problem of study heterogeneity, low-quality research and generalizability of study results. We hope this section helps readers understanding why this newly flourishing field lead to the production of such different researches

9. Consider discussing the limitations of the study, such as any potential biases in the included studies, the generalizability of the findings, and any other limitations that may affect the interpretation of the results.

A section “limitations” was added at the end of the discussion, which deals with the problem of study heterogeneity, low-quality research and generalizability of study results. We hope this section helps readers understanding why this newly flourishing field lead to the production of such different researches

10. Provide a clear conclusion that summarizes the main findings of the review and their implications for the field of orthopaedics. Additionally, consider suggesting future research directions based on the identified gaps and limitations.

Suggestions for future research were provided in the conclusions

11. Proofread the manuscript for grammar, punctuation, and formatting errors to ensure clarity and readability.

English language was revised by a native speaker expert in the field of immunology and cell biology.

Round 2

Reviewer 4 Report

Much improved 2nd draft.  

Minor English-language grammatical Editing still required.  

Reviewer 5 Report

Paper can be accepted.

Minor English grammar correction.